# Sugars and Organic Acids in 25 Strawberry Cultivars: Qualitative and Quantitative Evaluation

**DOI:** 10.3390/plants12122238

**Published:** 2023-06-07

**Authors:** Dragica Milosavljević, Vuk Maksimović, Jasminka Milivojević, Ilija Djekić, Bianca Wolf, Jan Zuber, Carla Vogt, Jelena Dragišić Maksimović

**Affiliations:** 1Institute for Multidisciplinary Research, University of Belgrade, 11030 Belgrade, Serbia; maxivuk@imsi.bg.ac.rs (V.M.); draxy@imsi.bg.ac.rs (J.D.M.); 2Faculty of Agriculture, University of Belgrade, 11030 Belgrade, Serbia; jasminka@agrif.bg.ac.rs (J.M.); idjekic@agrif.bg.ac.rs (I.D.); 3Institute for Analytical Chemistry, TU Bergakademie Freiberg, 09599 Freiberg, Germany; bianca.wolf@chemie.tu-freiberg.de (B.W.); jan.zuber@chemie.tu-freiberg.de (J.Z.); carla.vogt@chemie.tu-freiberg.de (C.V.)

**Keywords:** strawberry fruit, sugars, organic acids, mass spectrometry imaging, FT-ICR-MS, HPLC-ECD, HPLC-DAD, TQI

## Abstract

(1) The nutritional quality of strawberry (*Fragaria × ananassa* Duch) fruits, among others, is largely maintained by the presence of soluble sugars and organic acids. As the primary products of photosynthesis, they are energy depots in plants, necessary for the construction of cell constituents, but also serve as precursors of aromatic compounds and signaling molecules. (2) In this study, fruits of 25 strawberry cultivars were qualitatively and quantitatively characterized concerning individual sugars and organic acids by HPLC, FT-ICR-MS, and MS imaging analysis. In addition, the total quality index (TQI), as a novel mathematical model, was used to compare all individual parameters evaluated to obtain a quantitative single score, as an indicator of overall fruit quality. (3) Regardless of a large number of cultivars and monitored parameters that were studded, several cultivars stood out in terms of selected primary metabolites, such as ‘Rumba’, ‘Jeny’, and ‘Sandra’, while the latter had the best TQI score. (4) Intercultivar variations in sugars and organic acids profiles, along with other bioactive compounds, should be considered for selection of promising cultivars with improved naturally occurring nutraceutical traits. Besides the search for a pleasant taste, increased awareness of healthy nutrition resulted in heightening consumer demand for high-quality fruit.

## 1. Introduction

Among berry fruits, strawberry (*Fragaria* × *ananassa*, Duch.) is the most outspread and consumed fruit worldwide, the production of which has increased steadily in recent decades, reaching 9,175,384 t [1]. With an annual production of 22,427 t, Serbia is grouped among countries with low-intensive production. A substantial part of strawberry production is realized in open field on raised beds covered with black polyethylene foil. Cold-stored plants are used for summer planting with an average density from 40,000 to 44,440 plants per ha. Cultivars are regularly changed in the plantations aiming to introduce best-performing cultivars well adapted to local conditions, which offer high quality fruit in terms of sweetness, flesh firmness, attractive appearance, and long shelf-life. Most of the fruit produced is sold fresh on the local or foreign market, while a very small amount is frozen for later use and processing, making the agro-industrial strawberry chain important to the domestic economy.

The early ripening time makes strawberries popular among producers because of the quick financial return and among consumers due to a unique combination of attractive appearance, delicious taste, and nutritional value [2]. As a rich source of minerals, vitamins, folate, and phenolic compounds, as well as fibers, strawberry fruits are readily consumed, having an essential positive effect on the human diet and health [3,4].

The main indicators of strawberry fruit quality and taste are sugars and organic acids [5], which, along with the color, are the important attributes for consumer acceptance. The attractive color of fruit is shaped by sugar derivatives of anthocyanidins, which also brands sugars as the building blocks of important aesthetic components [6]. The predominant “sweet” metabolites in strawberry fruit are glucose, fructose, and sucrose, whereas the main “sour” metabolites are organic acids, especially citric and malic acids [7]. The strawberry flavor is governed by the balanced ratio between sugars and organic acids, which increases during the ripening [6]. After consumer assessment of external attributes, the decision for subsequent consummation is based on taste defined by internal quality parameters related to sugar-to-acid ratio, concentrations of bioactive compounds, and texture [8].

Nowadays, the routine qualitative and quantitative analysis of primary metabolites in fruits is based on modern separation techniques, such as liquid chromatography and/or mass spectrometry, while the investigation of their spatial distributions is limited by the accuracy of sampling and tissue differentiation techniques [9]. In that regard, mass spectrometry imaging (MSI) is an emerging technique that can be used for the simultaneous investigation of both content and spatial distribution of metabolites, without requiring expensive antibodies, staining, or complicated and time-consuming derivatization, pre-concentration, and purification of samples [2,10,11,12,13].

Since the quality and quantity of nutritive compounds in strawberry fruit are strictly associated with the fruit genotype [14], fruit breeding is constantly focused on the creation of new cultivars that meet increased consumer preferences regarding fruit quality. To benefit the health of consumers, the availability of new cultivars with both high sensorial and nutritional fruit qualities is the only option for the promotion of improved strawberry consumption.

Therefore, the aim of this study was to quantify and characterize selected primary metabolites in 25 strawberry cultivars, most of which are still in the testing phase. According to our knowledge, the spatial localization of these metabolites within the strawberry fruit has not been fully accomplished. Therefore, we employed high-performance liquid chromatography with diode array and pulsed amperometric detection (HPLC DAD and HPLC PAD), electrospray ionization Fourier transform ion cyclotron resonance mass spectrometry (ESI-FT-ICR-MS), and MSI analysis for the identification and quantitation of primary metabolites, which were successfully visualized by matrix-assisted laser desorption/ionization (MALDI). In addition, a novel mathematical model for calculating the total quality index (TQI) was used to assess all quality parameters that were evaluated regardless of the unit of expression and presented quantitatively as a single score, indicating the overall fruit quality. The intercultivar variation of sugars and organic acids profiles could be employed for the promotion of promising cultivars with improved fruit quality attributes that fulfill consumer demands, thus supporting the concept of an ‘educated consumer’.

## 2. Results and Discussion

### 2.1. HPLC-PAD Analysis of Sugars

Sensory experience in humans is principally driven by sugars and organic acids in a combined effect with volatile/aroma compounds providing the final perception [15]. Fruit taste quality is often related to internal attributes, primarily the concentration of individual sugars and their ratio. Thus, fruits with identical total sugar content which contain more fructose or sucrose taste sweeter than fruits with high glucose content [16].

In our study, in fruits of each cultivar three main soluble sugars, fructose, glucose, and sucrose, were identified and quantified by HPLC-PAD, with fructose being quantitatively the most abundant (Table 1), as also previously reported [17,18]. Compared to these studies, where the level of fructose and glucose was in the range of 19–43 and 13–26 g kg^−1^, respectively, their content was significantly higher in our cultivars. ‘Capri’, ‘Rumba’, and ‘Jeny’ cultivars stood out regarding the content of both fructose and glucose, while ‘Federica’ was predominant concerning sucrose content.

In all cultivars, reducing sugars (glucose and fructose) were contained in almost equal concentrations, while concentrations of sucrose were significantly lower, indicating that chemical or enzymatic hydrolysis occurred in some samples. Accordingly, the enhancement activity of invertase, a cell wall-bound enzyme responsible for sucrose hydrolysis to glucose and fructose, was found to be inversely proportional to sucrose concentration [19]. The same authors reported a drastic reduction of sucrose content in blueberry fruit during cold storage at −30 °C, which is assumed to be the case with our cold-stored samples since sucrose was not detected in some samples stored at −20 °C for a longer period (Table 1, n.d.).

Sweetness, as an internal quality trait of fresh fruit, is a desirable attribute that is usually governed by sugar content [16]. To calculate the sweetness of fruit, individual sugars’ quantification by instrumental assessment is a necessary step. In the estimation of the sweetness index, besides the content, the molar concentration of each sugar must be considered [17]. In our study, the sweetness index was the highest in ‘Rumba’, followed by ‘Jeny’, and ‘Sandra’ cultivars. The cultivar ranking for the sweetness index had a similar pattern as the cultivar ranking for the fructose and glucose content, indicating that monosaccharides are important indicators of fruit taste. Accordingly, fruit taste can be managed by preharvest conditions through changes in respiratory metabolism, in which glucose and fructose are the main substrates.

### 2.2. HPLC-DAD Analysis of Organic Acids

Besides sugars, the diversity and concentration of organic acids play an important role in strawberry organoleptic properties [20,21,22]. In many studies, citric acid was identified as the major organic acid in strawberry fruit regardless of cultivar or growing conditions [18,20,21,22]. In our study, besides citric acid, malic acid was also highly abundant, while shikimic and fumaric acids were quantified in trace amounts (Table 2). Quantified values of citric acid in our study were in the range of 3–7 mg g^−1^ FW, which is in accordance with the values reported for other strawberry cultivars [20,21,23]. Concerning the content of citric acid, ‘Laetitia’, ‘Sibilla’, and ‘Rumba’ were the leading cultivars, while ‘Capri’ and ‘Rumba’ were predominant in terms of malic acid. The content of shikimic acid was elevated in ‘Capri’, ‘Jeny’, and ‘Joly’, while ‘Sibilla’, ‘Premy’, and ‘Federica’ were rich in fumaric acid. Considering that taste attributes are driven primarily by citric and malic acid, where citric acid contributes to the sharp sour taste of fruit and malic acid contributes to the refreshingly tart taste [24], the balanced composition of both organic acids defines the perception of “tartness” and astringency in strawberry fruit.

Since the malate is less efficiently transported into the vacuole than citrate, fruit cells preferably store citrate, as respiratory substrate, rather than malate [25]. Hence, relatively small changes in the content of the main organic acids (citrate and malate) in fruit can be followed by their ratio. The lower the citrate-to-malate ratio, the more equal amounts of citrate and malate were stored in fruit. Contrarily, higher values may be caused by differences in vacuolar transport and/or in the rate of conversion of malate into citrate through the tricarboxylic acid cycle. As presented in Table 2, the citrate-to-malate ratio in all cultivars was in the range of 1.2–3.1, which is in line with the results reported for different strawberry cultivars grown in South Africa [20]. The highest values were recorded in ‘Lofty’, ‘Quicky’, and ‘Tea’ cultivars, implying that, in fruits of these cultivars, the TCA cycle is acting in a non-cyclic flux mode converting malate into citrate at a higher rate.

### 2.3. Total Quality Index (TQI)

TQI is a reliable mathematical tool that enables the comparison of quantitatively evaluated individual parameters of strawberry fruits, regardless of expression units, to obtain a single score for an overall fruit quality [26]. TQI results for all sugars and organic acids quantified by HPLC in fruits of 25 newly introduced strawberry cultivars are presented in Figure 1. Cultivar ‘Sandra’ had the best TQI score (1.29), followed by ‘Arosa’ (1.33), and ‘Irma’ (1.35). On the contrary, the worst TQI values were calculated for ‘Quicky’ (1.99), ‘Roxana’ (1.93), and ‘Sibilla’ (1.92). TQI showed that the overall fruit quality scores were clearly distinguished among different strawberry cultivars, while ‘Sandra’ could be considered the best cultivar in terms of overall fruit quality.

### 2.4. MS Imaging

Strawberries contain a complex mixture of different primary and secondary metabolites and biopolymeric molecules [27,28,29,30]. Hence, a comprehensive analysis of these biomarker compounds in a broad range of masses is only possible if ultrahigh-resolution mass spectrometric techniques, such as Fourier transform ion cyclotron resonance mass spectrometry (FT-ICR-MS), are applied. In general, FT-ICR-MS resolves the ionized analytes according to their individual cyclotron frequencies, which are inversely related to the *m*/*z* of the ions [31,32].

The characterization of biomarkers in the strawberry matrix is generally possible in two ways. One can either extract the different biomolecules from the strawberry fruit and analyze these extracts using, for example, electrospray ionization (ESI) or matrix-assisted laser desorption/ionization (MALDI) for ion generation. The latter approach makes it possible to study the overall composition of both polar and less polar biomarkers [33,34,35]. Secondly, the spatial distribution of biomarker compounds in the fruit body can be characterized by applying MSI, which enables immediate detection of compounds in tissue sections skipping extraction, separation, purification, or labeling procedures [36,37,38,39,40,41]. For this purpose, micro-sections of the plant parts are generated, usually with the help of a cryomicrotome. Afterwards, these tissue samples are overlayed with a photoactive matrix, which is necessary to produce ions of the biopolymer and biomarker molecules in the matrix-assisted laser desorption/ionization (MALDI) process. During the MSI analysis, the matrix-coated tissue sample is evenly divided into sample spots and at each spot a mass spectrum is recorded. From these mass spectral results, distribution maps of the organic compounds can be generated, which enable the verification of molecules in different plant or fruit organelles [2,10,11,12,13].

In order to study and visualize the spatial distribution of sugars, organic acids, and other biomarkers by means of FT-ICR-MS, cross and longitudinal sections of the cultivars ‘Aprika’ and ‘Sandra’ were prepared and analyzed by MSI. We opted for these two as representative cultivars for the whole sample set of strawberries: ‘Sandra’ as positive extreme in terms of TQI and ‘Aprika’ as average cultivar regarding all tested parameters.

Pictures of the tissue sections of both cultivars are presented in Figure 2. In addition, the spatial distribution of three potential sugar molecular ions and one organic acid ion are visualized in Figure 3. Judging from the assigned molecular formulae, we can suppose that the molecules of interest are a hexose (potentially glucose or fructose, Figure 3a–d), a disaccharide (potentially sucrose, Figure 3e–h), a trisaccharide (potentially maltotriose, Figure 3i–l), and a tricarboxylic acid (potentially citric acid, Figure 3m–p). All of these molecules were detected as [M + K]^+^ ions during the MSI analyses of the four tissue samples, which is attributable to the naturally high potassium content in strawberries [13]. Nonetheless, [M + H]^+^ and [M + Na]^+^ ions of these compounds were also found in all analyses, but to a significantly lower extent.

The general distribution trend is similar for all visualized molecules, as ions with comparable higher intensities were found for all four biomarkers in the skin, cortical tissue, and pith tissue region of the strawberry fruits, which is in good alignment with previously published data [2,10,11,12,13]. Mass spectral data from ‘Sandra’ showed higher numbers of spots, where an analyte of interest was detectable, in comparison to the data sets of ‘Aprika’. This distribution pattern of monosaccharides and disaccharides among cultivars corresponds with the sweetness index (Table 1), which was significantly higher in ‘Sandra’ compared to ‘Aprika’. Nonetheless, maximum intensity values for one biomarker molecule are in a comparable range, if the same type of tissue sample (cross or longitudinal section) is evaluated for both cultivars.

### 2.5. FT-ICR-MS Analysis of Extracts

In a further step to characterize biomarker molecules of higher molecular weight in the two representative strawberry cultivars ‘Aprika’ and ‘Sandra’, fruit extracts were generated and analyzed by ESI(−)- and ESI(+)-FT-ICR-MS. In order to study the general composition of these extracts, van Krevelen plots are utilized to visualize the variety of supposably contained compounds and compound classes in both extracts (Figure 4). For this visualization, an *O*/*C* ratio as well as an *H*/*C* ratio is calculated for each assigned molecular formula. Thus, each molecule is presented in this plot as a single point. Specific *O*/*C* and *H*/*C* regions can be correlated to different compound classes (lipids, sugars, peptides, lignins, etc.) to reveal the composition of a complex sample. In a van Krevelen plot, lignin structures can be found between *O*/*C* = 0.3–0.8 and *H*/*C* = 0.5–1.5. Condensed hydrocarbons occupy a similar *H*/*C* region, but typically possess lower values in the *O*/*C* region (0–0.3). Additionally, lipids (*O*/*C* = 0–0.3, *H*/*C* = 1.6–2.3), peptides/proteins (*O*/*C* = 0.3–0.5, *H*/*C* = 1.5–2.1), amino sugars (*O*/*C* = 0.5–0.8, *H*/*C* = 1.5–2.1), and carbohydrates (*O*/*C* = 0.8–1.3, *H*/*C* = 1.4–2.4) can be distinguished with the help of this plotting option [42].

According to Figure 4, compounds from all six compound classes are assumable in the extracts of cultivars ‘Aprika’ and ‘Sandra’. In particular, molecules which are lipids, peptides/proteins, and amino sugars are primarily detected by ESI(+). In contrast, ESI(−) seems to be more convenient for ionizing carbohydrates, as more data points are visible in the van Krevelen plot of the ESI(−)-MS data in this region (*O*/*C* = 0.8–1.3, *H*/*C* = 1.4–2.4). Thus, from these van Krevelen plots, we can derive that a high amount of organic acids, sugars, and polyphenolic compounds are present in both cultivars, as most of the ions were assigned to lipids, carbohydrates, lignin-like structures, and condensed hydrocarbons.

In order to learn more about the molecules and compound classes that are present in the extracts from ‘Aprika’ and ‘Sandra’, the mass spectral data were further evaluated by clustering the assigned molecular formulae into heteroatomic classes, based on the number of nitrogen, oxygen, and/or sulfur that was assigned to a molecular ion. These heteroatomic compound clusters were then evaluated according to the total number of assigned molecular formulae and the relative abundance of different heteroatomic classes. Visualizations for these two assessment parameters for compound classes N_1_O_4_–N_1_O_8_, S_1_, S_1_O_5_–S_1_O_6_ and O_1_–O_16_ are illustrated in Figure 5, as these compound classes showed the highest number of assigned molecular formulae.

From these plots it becomes obvious that the highest numbers of assigned molecular formulae from the illustrated compound classes were assigned to the ESI(+)-FT-ICR-MS data set of ‘Sandra’, even though that the amounts for ‘Aprika’ are in a comparable range. According to the ESI(−)-FT-ICR-MS data sets, slightly higher numbers are observable for ‘Aprika’. The relative abundancies are, in comparison, similar for both strawberry extracts for each individual heteroatomic class. In addition, we can derive from these plots that most of the compounds are assigned to oxygen-containing heteroatomic classes O_1_–O_16_. Most likely, these compounds are organic acids, esters, sugars, and phenolic compounds, which are typical compounds contained in strawberries [2,10,11,12,13,22,27,28,29,43,44]. Sulfur-containing esters or acids are furthermore imaginable for heteroatomic classes S_1_O_5_ and S_1_O_6_. Compounds of the heteroatomic classes N_1_O_4_–N_1_O_8_ were mainly assigned to the ESI(+)-FT-ICR-MS data sets of both cultivars. In accordance with the van Krevelen plots, these molecules could correspond to peptides/proteins and amino sugars.

The evaluation of the ESI-FT-ICR-MS data sets of both extracts was focused, in the next step, on the structural characterization of the different molecules in the various heteroatomic classes. For this purpose, visualizations of the carbon number (*n_C_*) vs. the double bond equivalent (*DBE*) of each heteroatomic class were utilized. Both *n_C_* and *DBE* are obtained using the molecular formula assigned to an ion. The number of assigned carbon atoms gives the *n_C_* value, whereas *DBE* can be calculated according to the number of assigned carbon, hydrogen, and nitrogen atoms. The *DBE* corresponds to the number of double bonds and cycles, which are present in the molecular structure. In contrast, *n_C_* values are used to assess the alkylation degree of a compound class [35].

The *n_C_*-*DBE* plots for heteroatomic classes O_4_–O_8_ are visualized in Figure 6. Furthermore, illustrations for heteroatomic classes O_1_–O_3_, O_9_–O_16_, N_1_O_4_–N_1_O_8_, as well as S_1_ and S_1_O_5_–S_1_O_6_, are illustrated in the Appendix A to this paper (see Appendix A). From these plots we can derive that mainly non-aromatic molecules with multiple hydroxy, carbonyl, aldehyde, carboxy or ester groups are potentially present in the heteroatomic classes O_1_–O_16_, as most of the compounds possess *DBE* values ≤ 5 (see Figure 6 and Appendix A). Thus, organic acids, such as oxocarboxylic acids (O_3_, *DBE* = 2), dicarboxylic acids (O_4_, *DBE* = 2), tricarboxylic acids (O_6_, *DBE* = 3), tetracarboxylic acids (O_8_, *DBE* = 4), pentacarboxylic acids (O_10_, *DBE* = 5), and hexacarboxylic acids (O_12_, *DBE* = 6) with *n_C_* values between 11 and 43 can be supposed in the extracts of both cultivars. Higher oxygen numbers and *DBE* values could correspond to analogous structures that contain additional oxygen-containing functional groups (e.g., hydroxy). Additionally, monosaccharides up to trisaccharides can be assumed according to the *n_C_*-*DBE* plots of the oxygen-containing heteroatomic classes, as molecules with a relatively high oxygen number (*o* ≤ 18) but low *n_C_* (*n_C_* ≤ 18) and *DBE* values (*DBE* ≤ 3) are present. Aromatic molecules are also assumable, according to the *n_C_*-*DBE* plots of compound classes O_6_–O_16_ (*DBE* ≥ 4). These molecules are, most likely, flavonoids or anthocyanins, which are part of the polyphenolic compounds that are typically found in strawberries [2,10,12,13,27,28,29,43,45].

Nitrogen- and oxygen-containing compounds were mainly assigned to the ESI(+)-FT-ICR-MS data sets of extracts from ‘Aprika’ and ‘Sandra’ (see Appendix A). Most of these compounds possess relatively high *DBE* values (*DBE* ≤ 17), which might correspond to aromatic and amino-functionalized molecules. In contrast, only small amounts of sulfur- as well as sulfur-/oxygen-containing compounds are present in the *n_C_*-*DBE* plots of compound classes S_1_ and S_1_O_5_–S_1_O_6_ (see Appendix A). Most of these compounds possess *DBE* values, which are attributable to the presence of aromatic sulfonates and sulfonic acids (*DBE* ≥ 5).

## 3. Materials and Methods

### 3.1. Chemicals

The chemicals for the sample extractions and for the mobile phases were HPLC-MS grade, methanol and phosphoric acid were purchased from Sigma-Aldrich (Burlington, MA, USA), while sodium hydroxide was purchased from J.T. Baker (Deventer, The Netherlands). HPLC grade carbohydrate standards (glucose, fructose, and sucrose), as well as citric and malic acid were procured from Fluka (Buchs, Switzerland), while shikimic and fumaric acid were purchased from Sigma-Aldrich (Burlington, MA, USA). All utilized solvents and chemicals for MS analysis were purchased from Merck Chemicals, Carl Roth or VWR Chemicals as HPLC (purity ≥ 99.9%) or analytical grade chemicals (purity ≥ 99%). Water for the mobile phase was double distilled and purified with the Milli-Q system (Millipore, Bedford, MA, USA).

### 3.2. Plant Material

Out of a total of 25 newly introduced strawberry cultivars that were used in our study, 22 are June-bearing cultivars (‘Clery’, ‘Alba’, ‘Joly’, ‘Aprika’, ‘Asia’, ‘Arosa’, ‘Roxana’, ‘Jeny’, ‘Laetitia’, ‘Garda’, ‘Lycia’, ‘Premy’, ‘Sibilla’, ‘Quicky’, ‘Federica’, ‘Lofty’, ‘Tea’, ‘Nadja’, ‘Arianna’, ‘Sandra’, ‘Vivaldi’, and ‘Rumba), while 3 are ever-bearing types (‘Albion’, ‘Capri’, and ‘Irma’). All cultivars were grown at the same plantation in Serbia located in the municipality of Šid (Serbia; 45°07′ N, 19°13′ E, 113 m a.s.l.), a region that is characterized by a temperate continental climate, with a mean annual air temperature of 10.7 °C and a mean annual precipitation of 650 mm. The soil was categorized as a fine sandy loam (pH 6.8) with medium to high levels of nutrients. Cold-stored plants of tested strawberry cultivars were planted on raised double beds covered with black polyethylene foil in July 2020 with a planting density of 44,440 plants per ha. Drip irrigation with two laterals per raised bed and emitters at a 10 cm distance were applied. Fertigation was performed at a frequency in accordance with crop requirements previously reported by Tomić et al. [46].

Fully ripe fruit samples were collected in three repetitions per 20 fruits (60 fruits per cultivar) during the second harvest in the first year after planting (2021). After harvest, fruits were stored at −20 °C prior to chemical analysis to prevent the effect of postharvest factors.

### 3.3. Sample Preparation

#### 3.3.1. Sample Preparation for HPLC Analysis

The whole fruits were carefully thawed, measured, and homogenized using mortar and pestle. Sugar compounds and organic acids were extracted in 80% methanol at a ratio of 1:3 (*w*/*v*). After centrifugation at 13,000× *g* for 10 min at 4 °C, supernatants were used for HPLC analysis of sugars and organic acids. Three extracts were prepared for each sample analyzed.

#### 3.3.2. Sample Preparation for ESI(+)- and ESI(−)-FT-ICR-MS Analysis

Two strawberry cultivars, ‘Aprika’ and ‘Sandra’, were selected as representative cultivars, for FT-ICR-MS and MSI analysis. ‘Sandra’ was chosen as the best-ranked cultivar concerning the TQI, while ‘Aprika’ was a middle-ranked cultivar regarding for all tested parameters. Differences in fruit quality traits among these two cultivars recommended them for comparative MS analysis.

Extracts of selected cultivars were prepared using adapted literature-known routines for the preparation of strawberry extracts [27,28]. Briefly, fruits were homogenized using pestle and mortar and 3 g of homogenate was transferred to an amber glass vial and 6 mL of an extraction solvent, consisting of 99% (*v*/*v*) acetone and 1% (*v*/*v*) acetic acid, was added. This suspension was sonicated for 15 min in the ultrasonic bath to extract organic compounds from the fruits. After this extraction time, the suspensions were filtrated through a cellulose filter (VWR, pore size 11 µm) and the filtrates were concentrated in a water bath at 37 °C until a final extract volume of 2 mL was reached. These final extracts were then diluted 20-fold using a solvent mixture consisting of 80% (*v*/*v*) methanol, 19% (*v*/*v*) dichloromethane, 0.9% (*v*/*v*) water, and 0.1% (*v*/*v*) triethylamine for ESI(−) analyses or with a solvent mixture that contained 60% (*v*/*v*) acetone, 30% (*v*/*v*) isopropanol, and 10% (*v*/*v*) water (+ 1 mM NH_4_COOCF_3_) for ESI(+) analyses.

### 3.4. HPLC Analysis of Sugars and Organic Acids

Separation of organic acids was performed on a Waters Breeze system (Waters, Milford, MA, USA) consisting of 1525 binary pump, thermostat, and 717+ autosampler connected to the Waters 2996 diode array detector adjusted at 210 nm. Supelco C-610H (300 × 7.8 mm) column (Sigma-Aldrich, Barcelona, Spain) was used, connected to the appropriate guard column. Isocratic elution was employed with 0.1% H_3_PO_4_ as the mobile phase at a flow rate of 0.5 mL min^−1^ and a column temperature of 40 °C. The data acquisition and spectral evaluation for peak confirmation were carried out by the Waters Empower 2 Software (Waters, Milford, MA, USA). Results were expressed as gram of specific organic acid per 100 g of fresh weight (g 100 g^−1^ FW). 

Sugar analyses were performed on the same Waters system connected to Waters 2465 electrochemical detector with 3 mm gold working electrode and hydrogen referent electrode. Separation of sugars was performed on CarboPac PA1 (Dionex, Sunnyvale, CA, USA) 250 × 4 mm column, equipped with corresponding CarboPac PA1 guard column at a constant temperature of 30 °C and a flow rate of 1.0 mL min^−1^. Sugars were eluted with 200 mM sodium hydroxide, prepared by dissolving a 10.5 mL of 50% *w*/*w*, low carbonate sodium hydroxide solution, (J.T. Baker, Deventer, The Netherlands), with vacuum degassed deionized water, to the final volume of 1 L. Signals were detected in the pulse mode within 150 ms of integration time using following waveform: E1 = +0.15 V for 400 ms; E2 = +0.75 V for 200 ms; E3 = −0.8 V for 200 ms. Filter timescale was 0.2 s, and the range was 1 μA for the full mV scale. The data acquisition and spectral evaluation for peak confirmation were carried out by Waters Empower 2 Software (Waters, Milford, MA, USA). Results were expressed as gram of specific sugar per 100 g of fresh weight (g 100 g^−1^ FW).

### 3.5. Total Quality Index (TQI) Calculation

All quality characteristics have been divided into two groups, in line with the works of Finotti et al. [47] and Djekic et al. [48]. Four organic acids (citric, malic, shikimic, and fumaric) were processed using rule #1 “the nearer to the target value, the better the quality”, Equation (1):(1)$$QI=\left|\frac{2*(x_{i}-T)}{x_{max}-x_{min}}\right|$$

Three other parameters (glucose, fructose, and sucrose) were processed in line with rule #2—“the higher the value, the better the quality”, Equation (2):(2)$$QI=\frac{\left|{x_{max}-x}_{i}\right|}{x_{max}-x_{min}}$$

For both equations, the following applies: *QI*—quality index for a specific quality characteristic; *x_i_*—measured value in the subset of values; *T*—target value (average value in the subset of values); *x_max_*—maximal value in the subset of values; *x_min_*—minimal value in the subset of values. 

Upon calculation of all *QI*s and considering all indices as vectors *QI* = (*QI*_1_, *QI*_2_, …, *QI_N_*) ϵ R^N^ [49], the total quality index (*TQI*) was calculated as per Equation (3) [47]:(3)$$TQI=\sqrt{\sum_{j=1}^{N}({{QI}_{j})}^{2}}$$

The rule of thumb for interpreting achieved *TQI* is “the lower the value, the better total quality index” [26].

### 3.6. Sweetness Index

The sweetness index was calculated by multiplying the sweetness coefficient of each sugar class (glucose = 1, fructose = 2.30. sucrose = 1.35) with individual sugar concentrations obtained by HPLC analysis, as described by Crespo et al. [17]. In this sweetness estimation approach, the contribution of each carbohydrate is based on the fact that fructose and sucrose are 2.30 and 1.35 times, respectively, sweeter than glucose.
sweetness index = (1.00 (glucose)) + (2.30 (fructose)) + (1.35 (sucrose))

### 3.7. FT-ICR-MS Analysis of Sugars and Organic Acids

Fruits of the strawberry cultivars ‘Aprika’ and ‘Sandra’ were characterized, as representative cultivars, by means of FT-ICR-MS. In a first step, cross and longitudinal sections of both cultivars were prepared to analyze the biomarker distribution with the help of MSI. For this purpose, the calyx of the strawberries was carefully removed and the samples were attached to a sample holder by embedding either the bottom part (cross sections) or a side (longitudinal sections) of the fruit with tissue freezing medium (Leica). Tissue sections with a thickness of 80 µm were generated using a Leica CM 1950 cryomicrotome. After every cut, the blade was cleaned with ethanol to avoid sample contaminations with the embedding medium. The sections were directly attached to glass slides, which were coated with indium tin oxide (ITO).

In a second step, the freshly prepared tissue sections were coated with 2,5-dihydroxybenzoic acid (DHB) (*c* = 30 g/L in acetone) using a self-made matrix sprayer. This sprayer consisted of an ICP nebulizer (Thermo Scientific Burgener Peek Mira Mist), which was linked to the nebulizer gas inlet and syringe port of the ESI source of the mass spectrometer. Thus, nebulizer gas pressure and matrix flow could be adjusted via the MS software. The distance between the tip of the sprayer and the sample was kept at 5 cm and the matrix was applied in five several spray cycles. Furthermore, a spraying pressure of 1.0 bar, 300 µL/min flow of matrix solution and 5 s spraying time per matrix layer were used. After each spray cycle, the coated tissue sample was oven-dried for 10 min at 30 °C [50].

### 3.8. Mass Spectrometry of Sugars and Organic Acids

MS experiments were conducted utilizing a 15 T solariX FT-ICR-MS from Bruker Daltonics, equipped with an ESI source and a Smart Beam II laser (frequency-tripled Nd:YAG laser, *λ* = 355 nm, pulse duration 3 ns, pulse energy 500 μJ, peak power 170 kW, average power 1.5 W) for MALDI-MS analyses. For the characterization of the fruit extracts of ‘Aprika’ and ‘Sandra’ cultivars, the ESI source was operated in positive and negative ion mode, using the following parameters: capillary voltage +2800 V (ESI(−))/−3000 V (ESI(+)), end plate offset −500 V, nebulizer pressure 1.0 bar, dry gas flow 4.0 L/min, dry gas temperature 300 °C, and syringe flow 10 µL/min. Mass spectra were acquired in a *m*/*z* range of 153.52–2000.00 Da by accumulating 256 scans. Resulting data sets had a size of 8 M, the resolving power was *R* = 800,000 at *m*/*z* = 400 Da, and a Q1 mass of 170 Da was used [51].

For the MSI experiments, the cross and longitudinal sections of both strawberry cultivars were scanned using an Epson Perfection V500 Photo scanner and a picture resolution of 4800 dpi. Before the imaging run the offset was corrected, if necessary, and the raster step size was set to 250 µm. In order to reduce the size of the MSI data sets, the MS data set size was decreased from 8 M to 2 M, which allowed the characterization of roughly 3000 sample spots in one hour. For all imaging analyses, the MALDI source was operated in positive ion mode. Furthermore, an ultra-large laser focus, a laser shot number of 20, a laser power of 27–31%, a plate offset, and a deflector plate voltage of 40 V and 200 V as well as a laser frequency of 500 Hz were applied. The other mass spectral parameters were kept analogous to the ESI-FT-ICR-MS analyses of the extracts.

Peak picking, calibration, and molecular formula assignment were accomplished using Bruker Daltonics software DataAnalysis 5.0 (SR 1). Mass spectral signals were considered peaks if their signal-to-noise-ratio (*s*/*n*) was ≥5. Mass calibration of the ESI-FT-ICR-MS experiments was conducted in a two-step process. In the first step, existing calibration lists were used for a first internal calibration. From these mass spectra, molecular formulae were calculated, and the resulting molecular formula lists were used to create an evolved calibration list, which contained molecular ions that are typical for the analyzed strawberry samples. Molecular formulae were assigned to peaks with *s*/*n* ≥ 10. The deviation from the theoretical masses should not exceed 0.3 ppm and the molecular formulae should show a predefined elemental composition (C*_c_*H*_h_*N*_n_*O*_o_*S*_s_*Na*_na_*K*_k_*: *c* = unlimited, *h* = unlimited, 0 ≤ *n* ≤ 3, *o* = unlimited, 0 ≤ *s* ≤ 5, 0 ≤ *na* ≤ 3, 0 ≤ *k* ≤ 1). Exported peak and molecular formula lists were imported into MATLAB R2022b (Mathworks) and further processed and visualized using in-house programs for blank correction and molecular formula filtering. Filtering of the molecular formula lists was conducted by applying the rules established by Herzsprung et al. [52,53] (double bond equivalent (*DBE*) ≥ 0, 0.3 ≤ *H*/*C* ≤ 2.5, *O*/*C* ≤ 1.0, *N*/*C* ≤ 1.0, *S*/*C* ≤ 1.0). The MSI experiments were also evaluated with the help of self-developed MATLAB programs using picture files (.tif) of the sections, spot lists (.csv files), and .imzML files, which were exported from Bruker Daltonics software flexImaging 5.0 (Build 80).

### 3.9. Statistical Analysis

Statistical analysis was performed using SPSS (IBM, Armonk, NY, USA). To examine significant differences in mean values of analyzed parameters between strawberry cultivars, analysis of variance (ANOVA) was used, while the Duncan test provided post hoc analysis. The level of statistical significance was set at 0.05.

## 4. Conclusions

The chemical composition of fruit becomes an important selection criterion in fruit production and one of the main marketing factors defined by consumer preferences. This comparative study revealed that sugars and organic acids profiles in fruits of different strawberry cultivars varied among them, which could be of the utmost importance in the selection and promotion of the most suitable cultivars regarding fruit quality. Irrespective of the large number of cultivars and parameters that were monitored in this study, several cultivars were distinguished from others in terms of the selected primary metabolites: ‘Rumba’, ‘Jeny’, and ‘Sandra’. The latter was also the best ranking newly released cultivar regarding the overall fruit quality, according to TQI score. Since strawberry fruit quality is a complex concept that depends on both the physical and chemical attributes of fruit, the ‘Jeny’ cultivar has been suppressed due to the pink color of the fruit which is quite unacceptable to the consumers (Appendix A). On the other hand, ‘Rumba’, as an early season cultivar with medium red fruit color and good size, remains a commercially important cultivar. The significance of intercultivar variations in primary metabolites profiles should be considered in breeding technology to provide a selection of promising cultivars with improved nutritive traits. The usage of FT-ICR-MS technologies in mapping the distribution of primary metabolites can be of great help in revealing the nature and function of these compounds naturally occurring in strawberry fruit. Fruits rich in phytonutrients, in addition to appearance and taste, can meet increasing consumer demands for a “Super Fruit”. 

Congregated, all of the presented data provide additional value to the “strawberry story”. Producers and breeders can be assured that in addition to the traditional detailed, quantitative analysis of primary metabolites, TQI can be used as a reliable tool for electing promising cultivars among newly introduced ones. On the other hand, detailed studies by FT-ICR-MS provide novel information regarding strawberry fruit metabolic distribution, which extremely important for understanding fruit physiology processes, going deep into the structures that define taste, acceptance, and hence the fruit quality.

## Figures and Tables

**Figure 1 plants-12-02238-f001:**
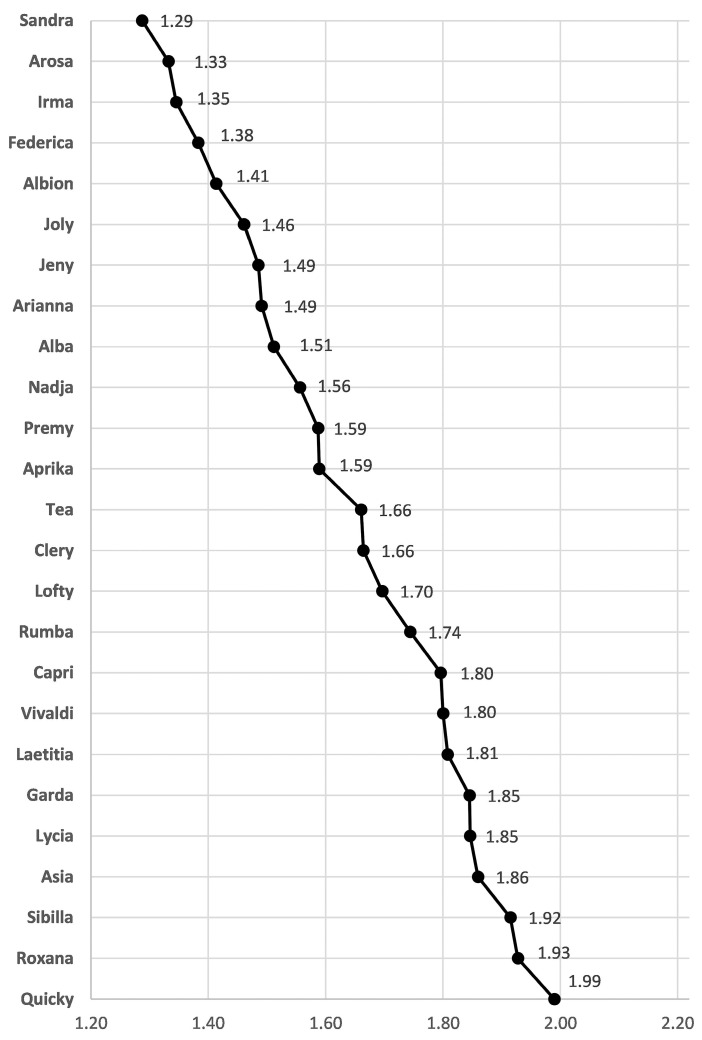
Total quality index (TQI) of the fruit extracts of different strawberry cultivars.

**Figure 2 plants-12-02238-f002:**
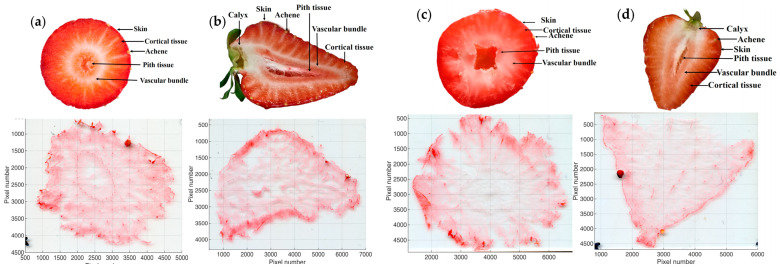
The tissue sections of two representative strawberry cultivars. (**a**) Cross-section of ‘Sandra’; (**b**) longitudinal-section of ‘Sandra’; (**c**) cross-section of ‘Aprika’; (**d**) longitudinal-section of ‘Aprika’.

**Figure 3 plants-12-02238-f003:**
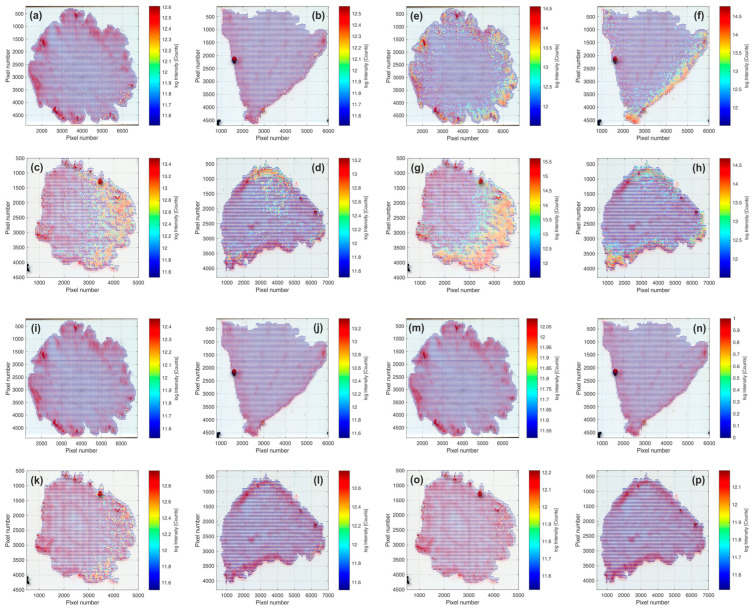
The distribution of biomarker molecules according to the MSI analyses of two representative strawberry cultivars. (**a**,**b**) C_6_H_12_O_6_ ([M + K]^+^, *m*/*z* = 219.026546 Da) in ‘Sandra’ and (**c**,**d**) in ‘Aprika’; (**e**,**f**) C_12_H_22_O_11_ ([M + K]^+^, *m*/*z* = 381.079369 Da) in ‘Sandra’ and (**g**,**h**) in ‘Aprika’; (**i**,**j**) C_18_H_32_O_16_ ([M + K]^+^, *m*/*z* = 543.132193 Da) in ‘Sandra’ and (**k**,**l**) in ‘Aprika’; (**m**,**n**) C_6_H_8_O_7_ ([M + K]^+^, *m*/*z* = 230.990161 Da) in ‘Sandra’, and (**o**,**p**) in ‘Aprika’. The observed intensity is presented logarithmically and color-coded (see color bar).

**Figure 4 plants-12-02238-f004:**
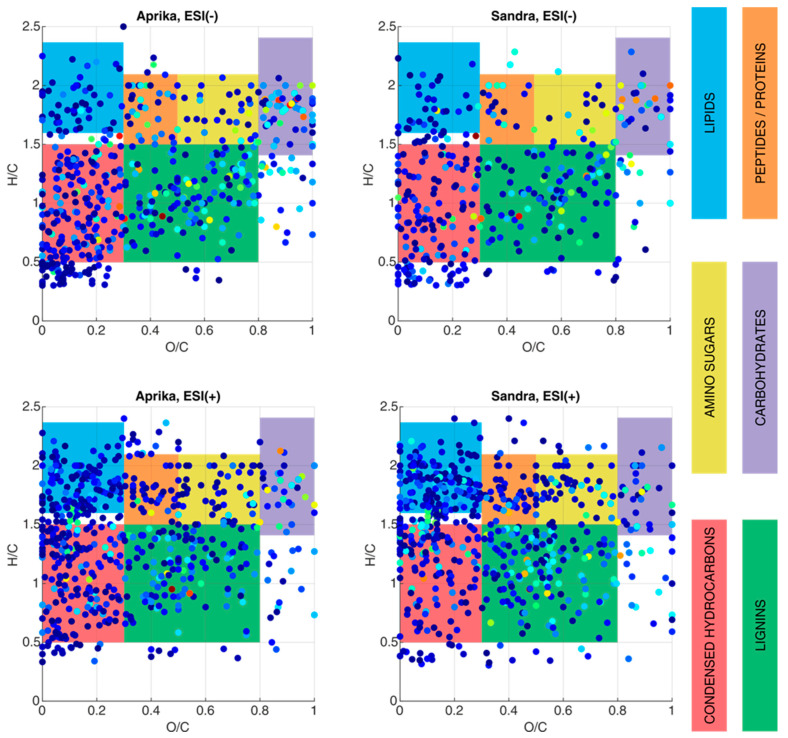
Van Krevelen plots, generated from the ESI-FT-ICR-MS data of extracts from two representative strawberry cultivars, ‘Aprika’ and ‘Sandra’. The compound classes were assigned according to Rivas–Ubach et al. [42]. The observed intensity is presented logarithmically and color-coded in the dots (blue: low intensity, yellow: medium intensity, red: high intensity). Substance classes are color-coded according to the description on the right side of the figure. ESI (−) data are given in the top row, ESI (+) data in the bottom row, respectively. Compounds that were not clustered into one of the highlighted compound classes belong to other compound classes, which were not evaluated for this research work.

**Figure 5 plants-12-02238-f005:**
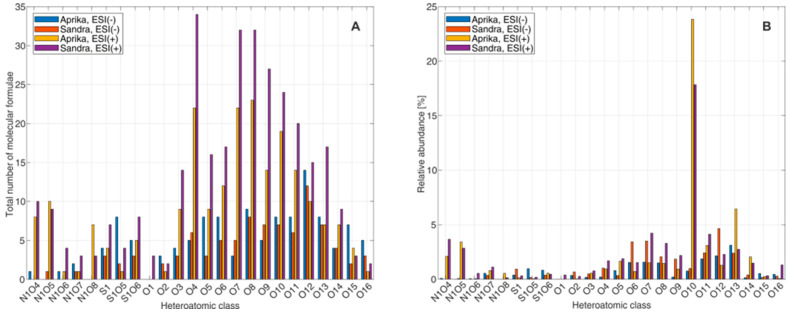
Comparison of assigned number of molecular formulae (**A**) and relative abundancies (**B**) to the FT-ICR-MS data sets of the strawberry extracts from two representative strawberry cultivars ‘Aprika’ and ‘Sandra’ for compound classes N_1_O_4_–N_1_O_8_, S_1_, S_1_O_5_–S_1_O_6_, and O_1_–O_16_.

**Figure 6 plants-12-02238-f006:**
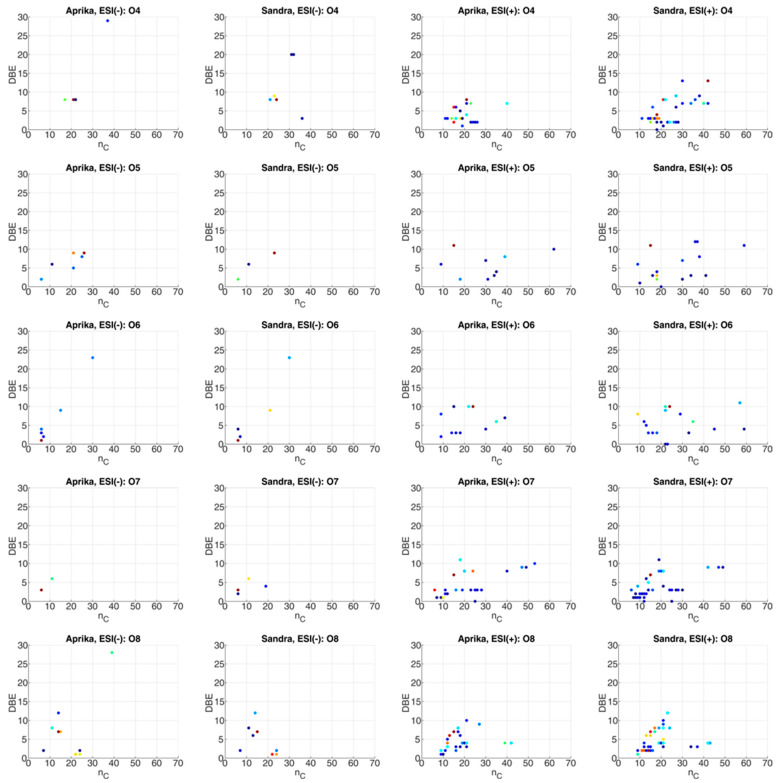
*n_C_*-*DBE* plots of compound classes O_4_–O_8_ for the extracts of representative strawberry cultivars ‘Aprika’ and ‘Sandra’, which were analyzed by ESI(−)- and ESI(+)-FT-ICR-MS. The observed intensity is presented logarithmically and color-coded (blue: low intensity, yellow: medium intensity, red: high intensity). In one row ESI(−) and ESI(+) data for one compound class are given for both cultivars.

**Table 1 plants-12-02238-t001:** Content (g 100 g^−1^ FW) of individual sugars and sweetness index (SI) in the fruit extracts of different strawberry cultivars.

Cultivars	Glucose	Fructose	Sucrose	SI
Roxana	1.73 ± 0.20 j	1.99 ± 0.34 h	n.d.	6.29 ± 0.96 l
Arosa	3.20 ± 0.32 b,c	3.65 ± 0.33 b,c,d	n.d.	11.58 ± 0.91 c,d,e,f,g
Joly	2.58 ± 0.21 c,d,e	2.72 ± 0.14 c,d,e,f	0.40 ± 0.02 f	10.23 ± 0.72 e,f,g,h,i,j
Asia	2.12 ± 0.12 g,h,i,j	2.44 ± 0.25 f,g,h	n.d.	7.73 ± 0.70 j,k,l
Alba	2.64 ± 0.33 c,d,e,f,g,h	2.97 ± 0.41 c,d,e,f	n.d.	9.54 ± 1.27 f,g,h,i,j,k
Aprika	2.70 ± 0.31 c,d,e,f,g	2.89 ± 0.53 c,d,e,f	0.78 ± 0.15 e	8.46 ± 0.09 i,j,k,l
Sibilla	2.17 ± 0.33 f,g,h,i,j	2.49 ± 0.23 f,g,h	0.80 ± 0.07 e	8.70 ± 0.83 h,i,j,k,l
Garda	2.08 ± 0.08 h,i,j	2.50 ± 0.17 f,g,h	n.d.	7.84 ± 0.46 i,j,k,l
Lycia	2.25 ± 0.26 e,f,g,h,i,j	2.39 ± 0.28 f,g,h	n.d.	7.75 ± 0.89 j,k,l
Jeny	4.34 ± 0.31 a	4.68 ± 0.40 a	n.d.	15.11 ± 0.81 a,b
Laetitia	3.52 ± 0.26 b	4.19 ± 0.46 b	n.d.	11.54 ± 1.33 c,d,e,f,g
Albion	2.62 ± 0.39 c,d,e,f,g,h	2.89 ± 0.48 c,d,e,f	n.d.	9.26 ± 0.36 g,h,i,j,k
Capri	4.84 ± 0.37 a	5.31 ± 0.38 a	n.d.	9.65 ± 0.89 f,g,h,i,j,k
Clery	2.22 ± 0.10 e,f,g,h,i,j	2.50 ± 0.24 f,g,h	n.d.	7.97 ± 0.66 i,j,k,l
Premy	2.69 ± 0.37 c,d,e,f,g,h	2.86 ± 0.36 d,e,f	0.82 ± 0.08 e	10.36 ± 1.10 e,f,g,h,i
Rumba	4.60 ± 0.22 a	4.79 ± 0.32 a	0.21 ± 0.06 f	15.91 ± 0.88 a
Vivaldi	1.95 ± 0.21 i,j	2.26 ± 0.25 g,h	n.d.	7.22 ± 0.89 k,l
Irma	3.18 ± 0.16 b,c,d	3.41 ± 0.09 b,c,d,e,f	n.d.	11.02 ± 0.35 d,e,f,g,h
Quicky	2.11 ± 0.10 g,h,i,j	2.32 ± 0.11 f,g,h	1.48 ± 0.13 c	9.44 ± 0.53 f,g,h,i,j,k
Nadja	2.63 ± 0.12 c,d,e,f,g,h	2.82 ± 0.12 d,e,f,g	2.11 ± 0.01 b	11.97 ± 0.42 c,d,e,f
Federica	2.77 ± 0.10 c,d,e,f,g	2.93 ± 0.12 c,d,e,f	3.12 ± 0.05 a	13.73 ± 0.44 a,b,c
Arianna	3.14 ± 0.22 b,c,d	3.53 ± 0.24 b,c,d	1.42 ± 0.09 c,d	13.18 ± 1.04 b,c,d
Lofty	2.52 ± 0.04 d,e,f,g,h,i	2.69 ± 0.01 e,f,g,h	1.16 ± 0.09 d	10.28 ± 0.23 e,f,g,h,i,j
Tea	2.91 ± 0.20 c,d	3.27 ± 0.21 b,c,d	1.65 ± 0.15 c	12.68 ± 0.88 c,d,e
Sandra	3.45 ± 0.14 b	3.72 ± 0.25 b,c	2.25 ± 0.18 b	15.04 ± 0.97 a,b

Data are presented as means (*n* = 3) ± standard error (SE). Values within a column with different lowercase letters are significantly different (*p* < 0.05), as determined using the Duncan comparison test. FW, fresh weight; n.d., not detected.

**Table 2 plants-12-02238-t002:** Content of individual organic acids (citric and malic acid, mg g^−1^ FW; shikimic and fumaric acid, μg g^−1^ FW) in the fruit extracts of different strawberry cultivars.

Cultivars	Citric Acid	Malic Acid	Shikimic Acid	Fumaric Acid	Citrate-to-Malate Ratio
Roxana	4.42 ± 1.07 e,f,g,h	3.02 ± 0.20 b,c	18.75 ± 0.42 b,c	7.70 ± 0.62 d,e	1.47
Arosa	4.48 ± 0.37 e,f,g,h	1.85 ± 0.15 f,g,h,i	17.00 ± 0.46 c,d	8.25 ± 0.26 c,d,e	2.43
Joly	4.86 ± 0.59 c,d,e	2.34 ± 0.32 d,e,f,g	19.84 ± 0.65 a,b	6.34 ± 0.31 e,f	2.10
Asia	3.52 ± 0.42 g,h,i	1.69 ± 0.17 g,h,i	18.65 ± 0.74 b,c	7.25 ± 0.38 d,e,f	2.08
Alba	4.21 ± 0.64 e,f,g,h	2.80 ± 0.45 b,c,d,e	18.00 ± 1.52 b,c,d	8.60 ± 0.57 b,c,d	1.50
Aprika	3.60 ± 0.27 g,h,i	2.33 ± 0.31 d,e,f,g	10.50 ± 0.68 i	9.30 ± 0.42 b,c	1.53
Sibilla	6.24 ± 0.31 a	2.70 ± 0.31 b,c,d,e,f	16.43 ± 0.31 d,e	13.05 ± 0.31 a	2.38
Garda	3.08 ± 0.31 j	2.62 ± 0.24 c,d,e,f,g	19.30 ± 0.74 b	7.80 ± 0.15 c,d,e,f	1.18
Lycia	3.45 ± 0.21 h,i,j	1.73 ± 0.16 g,h,i	12.70 ± 0.11 g,h	6.00 ± 0.15 e,f	2.00
Jeny	4.81 ± 0.51 c,d,e	2.93 ± 0.21 b,c,d	21.55 ± 1.56 a	6.50 ± 0.43 d,e	1.65
Laetitia	6.86 ± 0.43 a	2.76 ± 0.13 c,d,e,f	16.40 ± 0.64 d,e	8.30 ± 0.74 c,d,e	2.49
Albion	5.12 ± 0.33 b,c	2.88 ± 0.01 c,d,e,f	17.81 ± 1.17 c,d	8.78 ± 0.57 b,c,d	1.79
Capri	5.44 ± 0.37 b	3.73 ± 0.15 a	27.00 ± 1.48 a	8.48 ± 0.32 c,d,e	1.41
Clery	3.69 ± 0.11 g,h,i	2.06 ± 0.10 e,f,g,h	17.20 ± 0.61 c,d	7.80 ± 0.26 c,d,e,f	1.80
Premy	4.96 ± 0.72 c,d	2.84 ± 0.45 b,c,d,e	14.85 ± 1.17 f,g	11.45 ± 0.31 a	1.75
Rumba	5.93 ± 0.30 b	3.58 ± 0.17 a	15.35 ± 0.71 e,f,g	9.60 ± 0.30 b	1.65
Vivaldi	5.68 ± 0.08 b	2.63 ± 0.24 c,d,e,f,g	15.50 ± 0.95 d,e,f	9.35 ± 0.31 b,c	2.17
Irma	4.14 ± 0.09 f,g,h	1.92 ± 0.12 e,f,g,h,i	12.49 ± 0.54 g,h	9.49 ± 0.43 b	2.14
Quicky	4.56 ±0.03 e,f,g	1.59 ±0.01 h,i	1.95 ± 0.53 l	6.10 ± 0.42 e,f	2.88
Nadja	3.61 ± 0.11 g,h,i	3.14 ± 0.11 a,b	6.10 ± 0.11 j	7.40 ± 0.11 d,e,f	1.15
Federica	4.60 ± 0.37 e,f	2.32 ± 0.01 d,e,f,g	7.75 ± 0.11 i,j	11.45 ± 0.53 a	2.00
Arianna	4.51 ± 0.15 e,f,g,h	1.87 ± 0.10 f,g,h,i	5.60 ± 0.00 j,k	6.00 ± 0.11 e,f	2.20
Lofty	4.66 ± 0.11 e	1.50 ± 0.02 h,i	3.70 ± 0.42 k	7.50 ± 0.09 d,e,f	3.11
Tea	4.01 ± 0.09 f,g,h	1.47 ± 0.03 i	5.00 ± 0.53 k	4.70 ± 0.21 g	2.83
Sandra	3.59 ± 0.36 g,h,i	1.60 ± 0.11 g,h,i	5.70 ± 0.23 j,k	8.70 ± 0.11 b,c,d	2.19

Data are presented as means (*n* = 3) ± standard error (SE). Values within a column with different lowercase letters are significantly different (*p* < 0.05), as determined using the Duncan comparison test. FW, fresh weight.

## Data Availability

All new research data were presented in this contribution.

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
