# Peer review of "Sugars and Organic Acids in 25 Strawberry Cultivars: Qualitative and Quantitative Evaluation"

_plants, 2023, doi:10.3390/plants12122238_

Round 1
Reviewer 1 Report
The manuscript is well organized and the quality of results is good to be published in Plants.
I have comments to improve the quality of the manuscript.
-Title: Change selected primary metabolites by the families of metabolites studied: Sugars and organic acids.
-Abstract: The abstract have bullet points that should be removed. It is not clear why the authors just put numbers to the sentences. Please write the abstract as a resume of the results of the research as a traditional way.
-The point 4 in the abstract need to also include the presence of other bioactive compounds for example "Intercultivar variations in sugars and organic acids profiles together with the composition of other bioactives compounds should be considered...."
-The introduction should incorporate some background of the techniques used for metabolites qualitative screening.
-Line 76: Line 76: Any HPLC need a detector to measure something. Just write the detector/s or write Determination of sugars.
-Line 77. This sentence is out of context. Sensory experience also include volatile descriptors that are not related to sugars and organic acids. Take care of the concepts and the things that are written. Remove the sentence of write properly
-Line 82: add - HPLC-PAD
-line 83: change dominant by the most abundant, as was also ....
-Line 113: the same as title of section 2.1
-Line 118: acids
Remove in almost
Change detected by quantified. If you inform a concentration you were able to quantify. detection is normally used as an analytical terms to say that you find a compound, in a more qualitative way.
-Line 212: comparable
-Line 229: There is also possible to evaluate the presence of phenolic compounds? They are also important for the bioactive properties of strawberries.
-Lines 395-396: information of preparation of buffers is not needed.
The English need to be improved. I put some comments on the top, but a deep general revision of writting need to be performed.
Author Response
The manuscript is well organized and the quality of results is good to be published in Plants.
I have comments to improve the quality of the manuscript.
-Title: Change selected primary metabolites by the families of metabolites studied: Sugars and organic acids.
According to Reviewer’s recommendation, we changed the title into “Sugars and organic acids in 25 strawberry cultivars: qualitative and quantitative evaluation”.
-Abstract: The abstract have bullet points that should be removed. It is not clear why the authors just put numbers to the sentences. Please write the abstract as a resume of the results of the research as a traditional way.
During the manuscript preparation process, we followed the Instructions for Authors where the use of the Microsoft Word template is advised. In that template, the Abstract is divided into bullet points that are defined as follows: (1) Background, (2) Methods, (3) Results, and (4) Conclusions. We have tried to comply with the journal's requirements for manuscript preparation.
-The point 4 in the abstract need to also include the presence of other bioactive compounds for example "Intercultivar variations in sugars and organic acids profiles together with the composition of other bioactives compounds should be considered...."
We have changed point 4 according to your suggestion to “Intercultivar variations in sugars and organic acids profiles, along with other bioactive compounds, should be considered for selection of promising cultivars…”
-The introduction should incorporate some background of the techniques used for metabolites qualitative screening.
A technique used for metabolites qualitative screening, such as mass spectrometry imaging, was described in the Introduction section, in Lines 61-67.
-Line 76: Line 76: Any HPLC need a detector to measure something. Just write the detector/s or write Determination of sugars.
We have changed subtitle 2.1 to “HPLC-PAD analysis of sugars”, as you suggested.
-Line 77. This sentence is out of context. Sensory experience also include volatile descriptors that are not related to sugars and organic acids. Take care of the concepts and the things that are written. Remove the sentence of write properly
We used the term “principally” to avoid the wrong concept, but we added a brief explanation, as suggested: “Sensory experience in humans is principally driven by sugars and organic acids in a combined effect with volatile/aroma compounds providing the final perception.” (Lines 92-93)
-Line 82: add - HPLC-PAD
We have changed “HPLC PAD” to “HPLC-PAD” in Line 98, as you recommended.
-line 83: change dominant by the most abundant, as was also ....
We did it accordingly (Line 99).
-Line 113: the same as title of section 2.1
We have changed subtitle 2.2 to “HPLC-DAD analysis of organic acids”, as you suggested.
-Line 118: acids
We used the plural, as you recommended (Line 136).
Remove in almost
We removed “almost”, according to your suggestion.
Change detected by quantified. If you inform a concentration you were able to quantify. detection is normally used as an analytical terms to say that you find a compound, in a more qualitative way.
According to your suggestion, we replaced “detected” with “quantified” (Lines 136, 137, and 170).
-Line 212: comparable
We have changed “comparably” to “comparable” (Line 234).
-Line 229: There is also possible to evaluate the presence of phenolic compounds? They are also important for the bioactive properties of strawberries.
Indeed, they are, but in this paper, we choose to remain focused on primary metabolites. In our previous studies, cited as Refs 22, 43, and 45, we reported the profile and content of phenolic compounds in strawberry fruit. Phenolic compounds are very diverse secondary metabolites, so the results of such analysis/modeling require additional investigation. Also, polyphenols will take all of the reader’s attention mimicking our main goal to present a study on sugars and organic acids as the main attractiveness traits in strawberry.
-Lines 395-396: information of preparation of buffers is not needed.
In fact, it is. Although sounding trivial, if it is not strictly suggested to use low carbonate NaOH, the results of PAD analysis will be drastically compromised (Line 425).
Reviewer 2 Report
The article is well written and demonstrates the work carried out by the authors. However, I suggest making some improvements before its publication, such as:
Removing lines 24-25 from the abstract that refer to "super fruits" as it may confuse readers and future authors.
In the introduction, I noticed a lack of context regarding the strawberry cultivation and market in Serbia. Please add that information.
Lines 46-52, 381-385, lines 435-443, lines 454-493. Please provide the relevant references.
In the methodology, specify the number of fruits used in line 357.
The text is well written, and the English appear appropriate to me.
Author Response
The article is well written and demonstrates the work carried out by the authors. However, I suggest making some improvements before its publication, such as:
Removing lines 24-25 from the abstract that refer to "super fruits" as it may confuse readers and future authors.
According to the Reviewer’s suggestion, we removed "super fruits" and changed to “high-quality fruit”.
In the introduction, I noticed a lack of context regarding the strawberry cultivation and market in Serbia. Please add that information.
Following the Reviewer’s suggestion, we add some information about strawberry cultivation and the market in Serbia at the beginning of the Introduction section (Lines 30-44).
Lines 46-52, 381-385, lines 435-443, lines 454-493. Please provide the relevant references.
As Reviewer suggested, we provided relevant references to support our statements about MS imaging in the Introduction (Line 68), while mass spectral parameters for the characterization of strawberry extracts were analogous to previous work that we additionally cited in the M&M section (Lines 487, 500).
In the methodology, specify the number of fruits used in line 357.
The number of fruits used in the experiment was already defined in the second paragraph of the subsection “4.2. Plant material” (Lines 378-381).
Comments on the Quality of English Language
The text is well written, and the English appear appropriate to me.
Reviewer 3 Report
The paper of Milosavljević al. “Selected Primary Metabolites in 25 Strawberry Cultivars: Qualitative and Quantitative Evaluation” aimed to study primary metabolites in 25 new and/or rarely used strawberry cultivars. Generally, the paper includes a few scientific information (that can be described as a new) and a lot of around-scientific ways to analyze MS-data with questionable importance. The reference drawn up not in accordance with Instructions for Authors and should be corrected.
Highlights and strengths of the manuscript are:
The results may further increase interest in new strawberry cultivars and better understanding of strawberry metabolome.
Specific comments and suggested revisions:
Introduction section includes general information about strawberry chemical indicators and possibility of mass spectrometry as an instrument of analysis. I didn't understand why authors chose this object and what was the need of this work. Advantages and disadvantages of known methods of strawberry analysis should be disclosed, otherwise, it will remain incomprehensible is your study actual.
Results section contains the only new information about content of glucose, fructose, sucrose, citric acid, malic acid, shikimic acid, and fumaric acid in 25 cultivars obtained as a result of routine HPLC-procedure. Based on these data the new parameter, total quality index (TQI), was offered to characterize strawberry quality. But then the question arises about the need of TQI. In fact, it is another parameter describes sugar-to-acid ratio without any advantages over others. The sequence built on the basis of TQI values will be same as the sequence used ratio (total sugar content): (total acid content). All of this may look like a way to create another unnecessary method to describe strawberry quality.
The results of MS imaging did not seem to add anything useful for the strawberry metabolomics study and authors themselves confirm that the results are “in good alignment with previously published data [1,37-40]”.
The results of Van Krevelen diagram-involved study demonstrated the present not only primary metabolites but secondary too (which is logical). Finally, conclusions are drawn in relation to the FT-ICR-MS analysis are only that cultivars ´Aprika´ and ´Sandra´ has close metabolites with slight differences.
As a results, the paper is not looks like a well-structured and reasonable study and it needs to including additional data. There are numerous articles aimed to study of primary strawberry metabolites (especially sugar-acid-based studies) that must be taken into consideration. With all due respect to authors, I can’t recommend paper for publication due to the low novelty.
Minor editing of English language required
Author Response
The paper of Milosavljević al. “Selected Primary Metabolites in 25 Strawberry Cultivars: Qualitative and Quantitative Evaluation” aimed to study primary metabolites in 25 new and/or rarely used strawberry cultivars. Generally, the paper includes a few scientific information (that can be described as a new) and a lot of around-scientific ways to analyze MS-data with questionable importance. The reference drawn up not in accordance with Instructions for Authors and should be corrected.
According to the Instructions for Authors on the journal’s website, there are no strict formatting requirements: “Your references may be in any style, provided that you use the consistent formatting throughout. It is essential to include author(s) name(s), journal or book title, article or chapter title (where required), year of publication, volume and issue (where appropriate) and pagination. DOI numbers (Digital Object Identifier) are not mandatory but highly encouraged. The bibliography software package EndNote, Zotero, Mendeley, Reference Manager are recommended.” Thus, we used the EndNote Output Style template for MDPI ACS journals downloaded from the “MDPI Reference List and Citations Style Guide” to obtain the uniformity of references style.
Highlights and strengths of the manuscript are:
The results may further increase interest in new strawberry cultivars and better understanding of strawberry metabolome.
Specific comments and suggested revisions:
Introduction section includes general information about strawberry chemical indicators and possibility of mass spectrometry as an instrument of analysis. I didn't understand why authors chose this object and what was the need of this work. Advantages and disadvantages of known methods of strawberry analysis should be disclosed, otherwise, it will remain incomprehensible is your study actual.
As the first level of novelty of our study, we opted for a large number of newly introduced strawberry cultivars, of which many are still in the testing phase and there is no literature data about their content of primary metabolites. The second novelty level refers to methods that were used in our investigation which are state-of-the-art methods. Some of them were already utilized for examination of other strawberry cultivars, but the combination of all methods that were employed in our study is unique in the literature, based on our knowledge. We think that the joint engagement of all of these highly sophisticated methods brings very important information about the profile of primary metabolites among various strawberry cultivars. Thus, our comprehensive study would provide fundamental information related to sugars and organic acids qualitative and quantitative inter-cultivar variation that could be an important selection criterion for the identification of high-performing strawberry cultivars.
Results section contains the only new information about content of glucose, fructose, sucrose, citric acid, malic acid, shikimic acid, and fumaric acid in 25 cultivars obtained as a result of routine HPLC-procedure. Based on these data the new parameter, total quality index (TQI), was offered to characterize strawberry quality. But then the question arises about the need of TQI. In fact, it is another parameter describes sugar-to-acid ratio without any advantages over others. The sequence built on the basis of TQI values will be same as the sequence used ratio (total sugar content): (total acid content). All of this may look like a way to create another unnecessary method to describe strawberry quality.
Thank you for this comment, however, the data do not confirm your statement. As explained, the TQI is derived based on QI for the seven selected parameters - four organic acids and three sugars compounds. Total TQI showed that the top 3 were ´Sandra´, ´Arosa´, and ´Irma´. However, when each parameter is analyzed per se, we have the following top 3: citric (´Arosa´, ´Arianna´, ´Capri´), malic (´Aprica´, ´Federica´, ´Sibilla´), shikimic (´Irma´, ´Lycia´, ´Sibilla´), fumaric (´Albion´, ´Arosa´, ´Laetitia´), glucose (´Rumba´, ´Jenny´, ´Sandra´), fructose (´Rumba´, ´Jenny´, ´Sandra´), sucrose (´Federica´, ´Sandra´, ´Nadja´).
In addition, in order to check your statement that “The sequence built on the basis of TQI values will be same as the sequence used ratio (total sugar content): (total acid content)”, we calculated the ratio “total sugar content: total acid content” and the sequence was not the same as TQI. Top 3 cultivars were ´Tea´, ´Sandra´, and ´Arianna´ sorted by descending order, which is totally different than top 3 cultivars obtained by TQI: ´Sandra´, ´Arosa´, and ´Irma. As we explained in the paper, for TQI calculation quality characteristics have been divided into two groups:
#1 for organic acids - “the nearer to the target value, the better the quality” and
#2 for sugars – “the higher the value, the better the quality”
Since the sensorial quality of strawberry fruit is defined by the balanced content of sugars and acids, their simple ratio could not be a sufficient indicator of fruit quality.
Therefore, the calculation of a single TQI is advisable to select the best cultivar, based on the selected quality parameters defined as the main indicators of strawberry fruit quality and taste.
The results of MS imaging did not seem to add anything useful for the strawberry metabolomics study and authors themselves confirm that the results are “in good alignment with previously published data [1,37-40]”.
We cannot agree with the statement of the Reviewer that MS imaging did not add anything useful to the strawberry metabolomics study. With the help of the MS imaging experiments, we were able to characterize and visualize the spatial distribution of the most important sugars and organic acids in a strawberry fruit of cultivars that were not previously analyzed in such a way. All of our other analyses (HPLC, TQI, and ESI-MS) more or less give us a cumulative insight into these primary metabolic compounds with the employment of extensive extraction procedures, which were avoided in obtaining this information using MSI. The combination of results originating from both extracts and the fresh fruit itself helps us to understand not only what metabolomic compounds in which quantity are present, but also in which parts of the strawberry. Thus, the MSI results are fundamentally important for a comprehensive understanding of the biochemical composition of strawberries.
The fact that our findings by MSI are in good alignment with the previously published data from other research groups could be considered affirmative since this is a common way of comparing results in the literature. Therefore, it only confirms that the presented data are trustworthy and eligible to be used for a description of the metabolomic compounds of tested strawberry cultivars.
The results of Van Krevelen diagram-involved study demonstrated the present not only primary metabolites but secondary too (which is logical). Finally, conclusions are drawn in relation to the FT-ICR-MS analysis are only that cultivars ´Aprika´ and ´Sandra´ has close metabolites with slight differences.
We agree with the reviewer that the ion mode specific van Krevelen plots of both cultivars (“Aprika” and “Sandra”) look rather similar, even though slight differences are obvious. For instance, the number of data points which can be attributed to the different, assigned compound classes significantly varies between the data sets of both cultivars. Nonetheless, van Krevelen plots can only be used to visualize, in general, the chemical composition of a complex biochemical sample. To study and illustrate the differences between the ESI-FT-ICR-MS data sets of both extracted cultivars, we gave a detailed description and discussion of the amount of assigned heteroatomic classes (see Figure 5) as well as detailed structural suggestions based on the nC-DBE plots (see Figure 6 and Figures S1 – S5) in the subsequent section of the manuscript. Especially from the nC-DBE plots, it becomes obvious that minor but noticeable differences between both cultivars are detectable. For instance, the alkyl chain length (see nC values) for one given DBE drastically differs in the plots of compound classes O4, O7, or O8 (e.g., ESI(+), O4, DBE = 3 or ESI(+), O8, DBE = 2), if we evaluate the data sets of “Aprika” and “Sandra”.
In this section, we mainly focused on the interpretation of the data regarding the structural composition of the metabolomic compounds. We think that this information more accurately illustrates the general necessity and the value of the FT-ICR-MS data for the reader.
As a results, the paper is not looks like a well-structured and reasonable study and it needs to including additional data. There are numerous articles aimed to study of primary strawberry metabolites (especially sugar-acid-based studies) that must be taken into consideration. With all due respect to authors, I can’t recommend paper for publication due to the low novelty.
As we already justified, we think that the content of the manuscript is novel and significant due to the detailed investigation of primary metabolites of newly introduced strawberry cultivars, most of which are still in the testing phase. Diversiform information about sugars and organic acids consisted in strawberry fruit was obtained by state-of-the-art analytical methods, such as HPLC, FT-ICR-MS, and MS imaging analysis. In addition, the total quality index, as a novel mathematical model, was used to compare obtained individual parameters to a quantitative single score, as an indicator of overall fruit quality. Regarding the lack of literature data about the use of such combinations of methods for the comparison of primary metabolites among various cultivars, our comprehensive study would provide important information on sugars and organic acids qualitative and quantitative inter-cultivar diversity. This could be a significant selection criterion for the recognition of high-performing cultivars.
During the preparation of the original version of the manuscript, we carefully searched the literature and tried to include all relevant references in the manuscript. In this revised version we included several new due to the reviewers’ suggestions.
Comments on the Quality of English Language
Minor editing of English language required
Reviewer 4 Report
Authors have reported the Selected Primary Metabolites in 25 Strawberry Cultivars: Qualitative and Quantitative Evaluation the paper is well writen my only concern if possible corelates the reults with HPTLC. authors may go through similiars types of paper 10.56042/ijtk.v21i4.32514 .
Author Response
Authors have reported the Selected Primary Metabolites in 25 Strawberry Cultivars: Qualitative and Quantitative Evaluation the paper is well writen my only concern if possible corelates the reults with HPTLC. authors may go through similiars types of paper 10.56042/ijtk.v21i4.32514 .
The Reviewer suggested to correlate our results with HPLC data obtained in the mentioned paper. We carefully went through it, but unfortunately couldn’t find any connection with our results. In the referred paper, the structure of polyphenolic constituents in the polyherbal ayurvedic formulation was investigated by HPLC. Thus, the only connection is the apparatus, but we used different setup and separation methods for the detection of different class of compounds.
Round 2
Reviewer 1 Report
The authors made the suggested changes and the manuscript should be accepted.
Reviewer 3 Report
After correction the paper of Milosavljević et al. looks much better. I agree with the arguments put forward by authors and can recommend the paper for publication in present form.